

# Fusing Transformer-XL with bi-directional recurrent networks for cyberbullying detection

Md. Mithun Hossain[1], Md. Shakil Hossain[1], Md. Shakhawat Hossain[1], M. Firoz Mridha[2], Mejdl Safran[3], Sultan Alfarhood[3] and Dunren Che[4]

[1] Department of Computer Science and Engineering, Bangladesh University of Business and Technology, Dhaka, Bangladesh
[2] Department of Computer Science, American International University—Bangladesh, Dhaka, Bangladesh
[3] Research Chair of Online Dialogue and Cultural Communication, Department of Computer Science, College of Computer and Information Sciences, King Saud University, Riyadh, Saudi Arabia
[4] Department of Electrical Engineering and Computer Science, Texas A&M University—Kingsville, Texas, United States

Corresponding author
Sultan Alfarhood, sultanf@ksu.edu.sa

## ABSTRACT

Identifying cyberbullying in languages other than English presents distinct difficulties owing to linguistic subtleties and scarcity of annotated datasets. This article presents a new method for identifying cyberbullying in Bengali text data using the Kaggle dataset. This strategy combines Transformer-Extra Large (XL) with bi-directional recurrent neural networks (BiGRU-BiLSTM). Extensive data preparation was performed, including data cleaning, data analysis, and label encoding. Upsampling methods were used to handle imbalanced classes, and data augmentation enhanced the training dataset. We carried out tokenization of the text using a pre-trained tokenizer to capture semantic representations accurately. The model we presented, Transformer-XL-bidirectional gated recurrent units (BiGRU)-bidirectional long short-term memory (BiLSTM), which is called Fusion Transformer-XL, surpassed the performance of the baseline models, attaining an accuracy of 98.17% and an F1-score of 98.18%. Local interpretable model-agnostic explanation (LIME) text explanations were used to understand the reasoning behind the model's choices and performed the cross-dataset evaluation of the model using the English dataset. This helped improve the clarity and reliability of the proposed method. Furthermore, implementing k-fold cross-validation ensures our model's robustness and adaptability across diverse data categories. The results of our study demonstrate the effectiveness of combining Transformer-XL with bi-directional recurrent networks for detecting cyberbullying in Bengali. This emphasizes the significance of using hybrid architectures to address intricate natural language processing problems in languages with limited resources. This study enhances the development of methods for detecting cyberbullying and opens up opportunities for additional investigation into language diversity and social media analytics.

# INTRODUCTION

The rapid proliferation of social media platforms has transformed how individuals interact, enabling instantaneous communication across diverse cultures and languages (*Reed, 2018*). However, this shift has also given rise to a significant concern: cyberbullying. Defined as the intentional use of electronic communication to harm others, cyberbullying can have devastating effects on mental health and social well-being (*Mong, 2020*). While much research has been conducted in English-speaking contexts, there remains a substantial gap in understanding and identifying cyberbullying in non-English languages, particularly Bengali.

Current research predominantly focuses on cyberbullying detection in English (*Mahmud et al., 2023*), employing various machine-learning techniques and natural language processing (NLP) methods (*Mozafari et al., 2024*). Studies have explored the efficacy of different algorithms, including support vector machines (SVM), decision trees, and deep learning models, in identifying abusive language patterns (*Abro et al., 2020*; *Arif, 2021*; *Akhter et al., 2022*). However, these approaches often struggle with the linguistic nuances unique to other languages, compounded by the scarcity of annotated datasets tailored for such studies. This gap in research highlights the necessity for more inclusive methodologies that address the complexities of cyberbullying across diverse linguistic and cultural contexts.

In this article, we present a novel approach for detecting cyberbullying in Bengali text data by leveraging advanced machine learning techniques. Specifically, we propose a Fusion Transformer-XL model, which integrates a Transformer-XL architecture with bidirectional gated recurrent units (BiGRU) and bidirectional long short-term memory (BiLSTM) networks. This hybrid approach harnesses the strengths of both global context awareness through the Transformer-XL and localized context extraction *via* BiGRU and BiLSTM, allowing for a more nuanced understanding of cyberbullying patterns.

The Transformer-XL architecture effectively understands the Bengali context. Our goal is to enhance its detection capabilities by integrating bidirectional recurrent networks, specifically BiGRU-BiLSTM, which are effective in capturing bidirectional contextual information. Utilizing a Kaggle dataset containing 44,001 Bengali comments, we trained and evaluated our proposed model in a real-world scenario. The dataset was partitioned into a 70:15:15 ratio for training, testing, and validation, and cross-dataset evaluations were conducted using an English dataset. The proposed model achieved a remarkable 98.18% weighted F1-score and 98.17% accuracy in detecting cyberbullying in Bengali. In summary, this study contributes the following:

- This study presents a new fusion model that integrates Transformer-XL with bidirectional recurrent networks (BiGRU-BiLSTM) to identify cyberbullying in Bengali. This fusion design exploits the advantages of both models, thereby improving the comprehension of contextual information and long-range connections in textual material.
- Extensive data preparation is performed, including data cleaning, analysis, and preprocessing methods such as label encoding, resolving class imbalance by upsampling,

and data augmentation. These preprocessing steps are essential for maintaining the quality and efficacy of the model training process.

- The fusion model was compared with several baseline designs-such as Transformer, Transformer-GRU, Transformer-LSTM, Transformer-GRU-LSTM, hybrid Transformer (Transformer-BiGRU-BiLSTM), and Transformer-XL. Our model surpasses existing baseline models in terms of accuracy and F1-score, showcasing its efficacy in detecting cyberbullying.
- We evaluated the performance of the Fusion Transformer-XL model using dataset in English, demonstrating the model's versatility and resilience in a variety of linguistic contexts.

Overall, this study demonstrates an approach combining transformer-XL with BiRNNs to improve cyberbullying identification in the Bengali language and to tackle intricate natural language processing problems in languages with limited resources. It is necessary to consider that the dataset may not encompass all types of cyberbullying, potentially capturing only specific forms or contexts while overlooking others. Additionally, it could be biased towards a particular time, reflecting the prevalent forms of cyberbullying during that era.

The rest of the article is structured as follows. Some of the related previous works are discussed in "Literature Review". "Problem Definitions" presents the problem definitions, and "Proposed Methodology: Fusion Transformation" presents the proposed methodology. "Experimental Setup" provides an overview of the experimental setup. Result analysis is discussed in "Result Analysis". Finally, in the "Discussion", "Limitation and Future Work" presents limitations and future work, followed by a conclusion in "Conclusions".

## LITERATURE REVIEW

The widespread use of the internet and smartphones has greatly amplified the population of social media users, resulting in a widespread occurrence of cyberbullying. The absence of strict legal frameworks and the ability to remain anonymous online worsens this problem, leading to significant psychological distress for the victims. Initial investigations into the identification of cyberbullying have used natural language processing (NLP) methodologies, machine learning (ML) algorithms, and deep learning (DL) models to tackle this issue. This literature review investigates the development of techniques for identifying cyberbullying, specifically examining the transition from conventional machine learning methods to more sophisticated deep learning models and pre-trained language models. The study focuses particularly on the detection of cyberbullying in Bengali.

### Machine learning approaches

*Hoque & Seddiqui (2023)* extensively explored machine learning approaches for detecting cyberbullying. In the study, after comparing models like SVM, random forest (RF), long short-term memory (LSTM), and bidirectional encoder representations from transformers (BERT) to identify cyberbullying in Bengali; the BERT-based model has the greatest accuracy of 80.165%. The study, which is sponsored by Bangladesh's ICT division,

emphasizes the requirement of feature selection and pre-processing as well as the need of further study in low-resource languages. Social media elements increased prediction performance for all evaluated machine learning (*Bozyiğit, Utku & Nasibov, 2021*) have shown to be important in cyberbullying detection. *Talpur & O'Sullivan (2020)* produced features from Twitter content in a binary setting, using a pointwise mutual information strategy in a cyberbullying detection framework. The findings were encouraging concerning Kappa, classifier accuracy, and f-measure metrics. *Cheng et al. (2020)* proposed an unsupervised cyberbullying detection model using a time-informed Gaussian mixture model that beat state-of-the-art unsupervised methods and obtained competitive performance relative to supervised models.

## Deep learning approaches

*Saifullah et al. (2024)* proposed BullyFilterNeT, which is a deep learning system designed for Bengali cyberbullying detection. According to the study, the BanglaBERT-based model achieves an astounding accuracy of 88.04%. In a similar vein, *Jobair et al. (2023)* investigate Bengali hate speech identification using LSTM, bidirectional long short-term memory (BiLSTM), gated recurrent unit (GRU), convolutional neural network (CNN), and BERT models. In the study, BERT shows 80% accuracy on a new dataset and 97% on an old dataset. These results highlight how well transformer-based deep learning models work to combat hate speech and cyberbullying in languages with limited resources. Moreover, *Kumar & Sachdeva (2022b)* integrates textual, visual, and info-graphic data to present a deep neural network for multimodal cyberbullying detection, which achieves remarkable performance with an area under the receiver operating characteristic (AUC–ROC) of 0.98. Finally, *Das et al. (2021)* uses an attention-based recurrent neural network to address hate speech detection in Bengali social media.

## Hybrid approaches

*Wahid & Al Imran (2023)* proposed the multi-feature transformer and deep neural network for Bengali multi-dimensional cyberbullying detection. The researchers combine user profiles, lexical characteristics, contextual embeddings, and semantic linkages for better efficiency of the model. *Iwendi et al. (2023)*, on the other hand, shows that deep learning architectures are effective in detecting cyberbullying. To improve cyberbullying detection in social media content, *Kumar & Sachdeva (2022a)* developed a Bi-GRU-Attention-CapsNet (Bi-GAC) model that uses sequential semantic representations and geographical location data. *Paul, Saha & Hasanuzzaman (2022)* also presents a deep learning-based multimodal framework and achieves encouraging results with a 0.75 F-measure on the Vine dataset by employing ResidualBiLSTM-RCNN architecture. In addition, *Karim et al. (2021)* provides DeepHateExplainer, an explainable model for Bengali hate speech identification that outperforms baselines from neural networks and conventional machine learning in identifying different kinds of hate speech.

## Transformer approaches

*Han et al. (2023)* describe the LTAnomaly model, which combines a Transformer with LSTM for efficient anomaly identification and represents logs using semantic information

and sequence links. *Teng & Varathan (2023)* evaluate conventional machine learning and transfer learning methods for cyberbullying detection by employing AMiCA data and methodical annotation procedures. *Roy & Mali (2022)* use transfer learning in text-based social media cyberbullying detection methods. *Elsafoury et al. (2021b)* examine attention weights and feature significance scores to clarify BERT's effectiveness in detecting cyberbullying throughout a range of datasets. *Elsafoury et al. (2021a)* further suggest employing modern contextual language models such as BERT and slang-based word embeddings to provide better representations of cyberbullying-related datasets.

Finally, a survey of current research in cyberbullying detection and deep learning approaches applied to social media data emphasizes the field's dynamic nature. This comprehensive analysis demonstrates that approaches for evaluating textual data from social media networks are always evolving and being refined. However, several obstacles remain, such as the need for larger and more diverse datasets, greater model interpretability, and increased algorithm scalability. These shortcomings provide an opportunity for future study into fresh techniques and solutions in cyberbullying detection.

The findings of this article provide the foundation for our proposed Fusion Transformer-XL architecture, which intends to contribute to bully categorization by utilizing advances in deep learning and cyberbullying detection approaches. We want to develop cyberbullying detection in social media situations by expanding on current knowledge and tackling highlighted limitations.

## PROBLEM DEFINITIONS

### Informal definition
Cyberbullying detection is the process of identifying and categorizing cyberbullying incidents in digital communication systems. It entails identifying written information on the internet that displays abusive, harassing, or offensive behavior towards people or organizations.

### Formal definition
The goal of cyberbullying detection is to create a system that can automatically recognize instances of cyberbullying in text data. The objective is to construct a predictive model $f(x_i)$ that can accurately assign labels to new, unseen text samples. The dataset $D$ contains text samples $x_i$ and their corresponding labels $y_i$, where $x_i$ represents a piece of text and $y_i$ denotes its cyberbullying classification label (*e.g.*, "Not bully," "Troll," "Religious," "Sexual," or "Threat"). The model's goal is to distinguish between different types of cyberbullying behavior by identifying patterns and characteristics that indicate the use of abusive or harassing language. This will aid in the development of more secure online settings.

## PROPOSED METHODOLOGY: FUSION TRANSFORMER-XL
Figure 1 shows the proposed Fusion Transformer-XL model architecture, which uses a combination of Transformer-XL, BiGRU, and BiLSTM networks to address the problem of

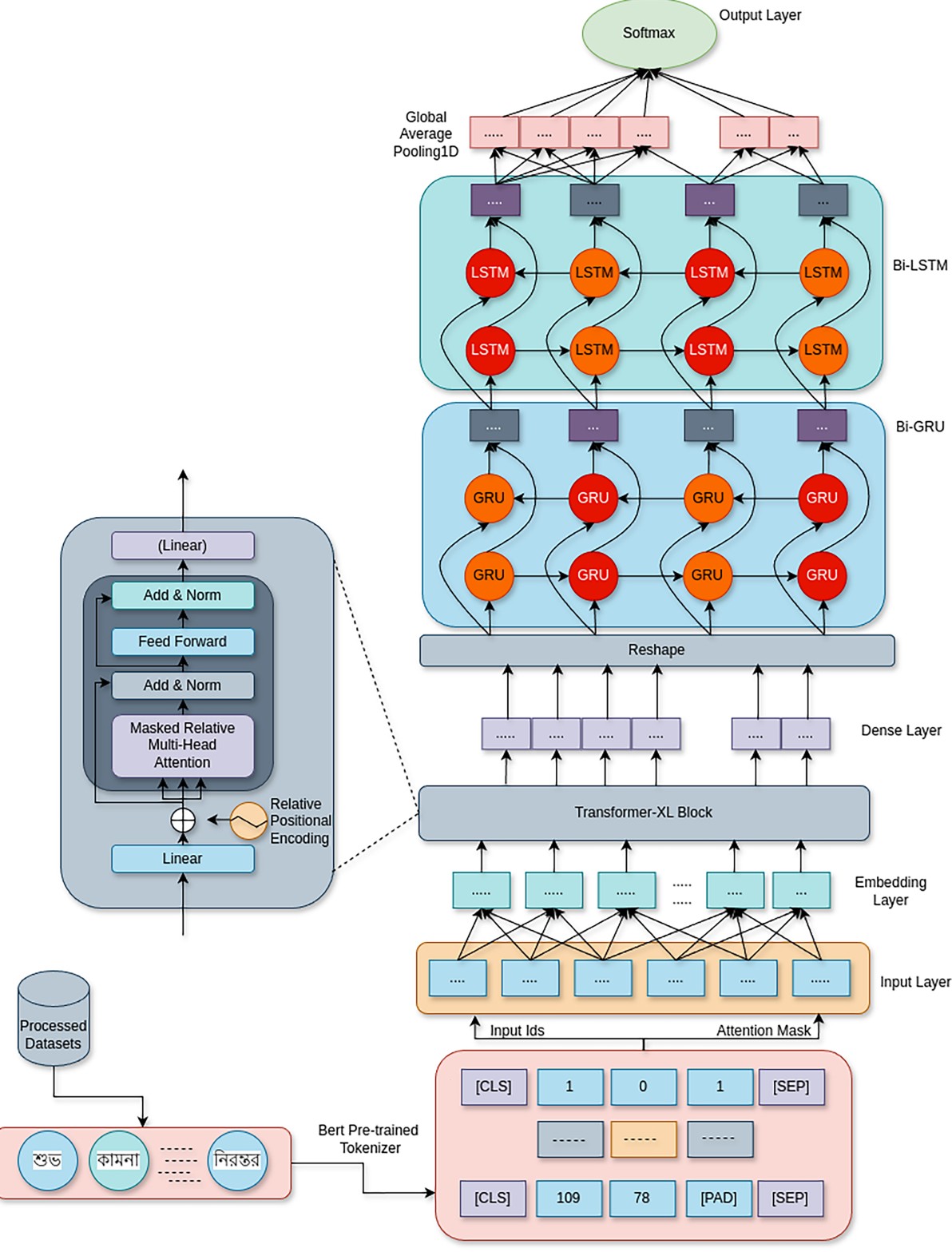

**Figure 1** This fusion architecture combines the strengths of Transformer-XL's long-range dependency handling with the temporal dynamics captured by bi-directional recurrent networks, creating a robust model for detecting various forms of cyberbullying with high accuracy.

cyberbullying detection. By combining the benefits of recurrent networks with attention-based processes, this hybrid design successfully extracts contextual information and long-range relationships from textual input.

1. **Input layer:** The model begins with two input layers:

   ○ **Input IDs:** This layer receives the tokenized form of the input text, where each token corresponds to a unique identifier in the vocabulary.

   ○ **Attention mask:** This layer is essential to the architecture of the Transformer. It facilitates the model's disregard for padding tokens or tokens that don't add to the text's meaning so that only pertinent tokens have an impact on attention estimations. The mathematical expression for the attention mask $M$ is as follows:

   $$M_{ij} = \begin{cases} 1 & \text{if } x_j \text{ is a valid token} \\ 0 & \text{if } x_j \text{ is a padding token.} \end{cases}$$

2. **Embedding layer:** The input IDs are converted into dense vectors of fixed size $d_{model}$ by the embedding layer. The representation of this transition is:

   $$E(x) = W_e \cdot x + b_e$$

   where $W_e$ is the embedding matrix, $b_e$ is the bias term, and $E(x)$ is the embedding of token $x$. In order to enable the model to capture semantic links between tokens, this phase transforms discrete token IDs into continuous representations. A dropout mechanism $D$ can also be incorporated into the embeddings to avoid overfitting during training:

   $$D(E(x)) = \begin{cases} E(x) & \text{with probability } p \\ 0 & \text{with probability } 1 - p. \end{cases}$$

3. **Transformer-XL blocks:** The model is made up of many Transformer-XL blocks. This architecture improves upon the Transformer standard by:

   ○ **Long-range dependency handling:** Long sequences may be difficult for traditional transformers to handle because of their fixed-length context windows. By adding relative positional encoding $PE_{rel}$, the Transformer-XL lessens this restriction and enables the model to represent connections between tokens based on their relative placements. Relative positional encoding in the attention process can be stated as follows:

   $$Attention(Q, K, V) = softmax\left(\frac{Q(K + PE_{rel})^T}{\sqrt{d_k}}\right)V.$$

   Because words may have drastically different meanings depending on their context, this is especially crucial for activities like detecting cyberbullying.

   ○ **Memory augmentation**: Transformer-XL can handle sequences longer than the input length since it has a technique for memory preservation across segments. The model's comprehension of context is improved by the memory $M$, which holds data

from earlier segments. The following is an expression of the memory update mathematical representation:

$$M_t = f(M_{t-1}, x_t)$$

where $M_t$ is the updated memory at time $t$, $M_{t-1}$ is the memory from the previous time step, and $x_t$ is the current input.

4. **Dense layer:** The concealed representations are processed through a fully linked dense layer after their passage through the Transformer-XL blocks. This layer functions as a step in between for the extraction of features; the transformation may be expressed as:

$$H = W_d \cdot X + b_d$$

where $H$ is the output, $W_d$ is the dense layer weight matrix, and $b_d$ is the bias vector.

5. **Bidirectional layers:** BiLSTM and bidirectional gated recurrent units (BiGRU) networks are used in the model. These elements are essential for:

   ○ **Contextual awareness:** The contextual representation of the input data is enhanced by the model's ability to extract context from both forward and backward sequences, a feature shared by both GRU and LSTM.

   ○ **Long-term dependencies:** Although long-term dependencies are naturally managed by GRUs and LSTMs, their bidirectional combination enhances the model's capacity to identify complex patterns. Mathematically speaking, the hidden states of GRU and LSTM are as follows:

$$h_t^{GRU} = f(h_{t-1}, x_t) \quad \text{and} \quad h_t^{LSTM} = g(h_{t-1}, x_t)$$

   where $h_t^{GRU}$ and $h_t^{LSTM}$ are the hidden states at time $t$ for GRU and LSTM, respectively.

6. **Global average pooling:** A global average pooling layer limits the output to a fixed-size vector after the bidirectional levels. The representations are averaged across the length of the sequence by this operation:

$$H_{pool} = \frac{1}{T} \sum_{t=1}^{T} h_t$$

where $h_t$ is the output from the preceding layer at time $t$, and $T$ is the length of the input sequence. This function provides a succinct overview of the acquired characteristics, particularly beneficial for categorization assignments.

7. **Output layer:** With a softmax activation function and a dense output layer, the model concludes. For the goal categories (cyberbullying and non-cyberbullying, for example), this layer converts the pooled representations into class probabilities. The resultant output can be shown as:

$$P(y_i \mid H) = \frac{e^{HW_{o_i}}}{\sum_{j=1}^{C} e^{HW_{o_j}}}$$

where $P(y_i \mid H)$ is the predicted probability of class $i$, $H$ is the pooled representation,

$W_{o_i}$ is the output weight vector for class $i$, and $C$ is the total number of classes. The model is trained using categorical cross-entropy loss:

$$L = - \sum_{i=1}^{C} y_i \log(P(y_i \mid H))$$

where $y_i$ is the true distribution (one-hot encoded).

To sum up, the Transformer-XL, BiGRU, and BiLSTM network strengths are successfully combined in the suggested hybrid model architecture for cyberbullying detection. The model utilizes the bidirectional capabilities of GRUs and LSTMs in conjunction with the sophisticated features of Transformer-XL for long-range dependency handling and memory retention. This allows the model to effectively capture the complex patterns and contextual subtleties found in textual data pertaining to cyberbullying.

### Proposed fusion Transformer-XL model training algorithm

Algorithm 1 displays the Fusion Transformer-XL model's training process, which is intended for the identification of cyberbullying. The input parameters, which include the input shape, vocabulary size, embedding dimension, number of attention heads, and dropout rate, among others, are defined first by the algorithm. To build the architecture, the model function initializes an embedding layer initially, then iteratively adds numerous Transformer-XL blocks to extract contextual information from the input data. To estimate class probabilities for the target categories, the output from the Transformer blocks is then processed *via* a sequence of Dense, Bidirectional GRU, and LSTM layers, ending with a softmax output layer. The implementation of a structured strategy enhances the model's capacity to identify intricate patterns within the data, hence improving its efficacy in cyberbullying detection assignments.

## EXPERIMENTAL SETUP

In our experimental setup, we focused on two distinct publicly available datasets annotated for cyberbullying detection: one in Bengali (*Ahmed et al., 2021*) and the other in English which is found data available section. The Bengali dataset includes five categories: Troll, Not Bully, Sexual, Religious, and Threat, which capture various forms of cyberbullying behaviors prevalent in Bengali social media contexts. Each category represents specific types of abusive language, enabling our model to learn nuanced patterns associated with cyberbullying. Conversely, the English dataset is structured around four categories: Ethnicity/Race, Not cyberbullying, Religion, and Gender/Sexual, reflecting the diverse nature of online harassment in English-speaking environments. By employing these two datasets, we aimed to create a comprehensive framework for detecting cyberbullying across different languages, facilitating a deeper understanding of the socio-cultural dynamics involved in online aggression. This approach not only enhances the model's adaptability but also underscores the importance of addressing cyberbullying in multilingual contexts.

**Algorithm 1** Fusion Transformer-XL model training algorithm.

1: **Input:**
2:     $input \leftarrow (120,)$     ▷ Input shape of the data
3:     $vocab \leftarrow tokenizer.vocab\_size + 1$     ▷ Vocabulary size including padding
4:     $d\_model \leftarrow 256$     ▷ Embedding dimension
5:     $num\_heads \leftarrow 2$     ▷ Number of attention heads
6:     $d\_ff \leftarrow 256$     ▷ Feed-forward dimension
7:     $num\_blocks \leftarrow 2$     ▷ Number of Transformer blocks
8:     $dropout \leftarrow 0.2$     ▷ Dropout rate
9:     $m\_len \leftarrow 128$     ▷ Memory length for Transformer-XL
10: **Output:**
11:     Trained Fusion Transformer-XL model
12: **function** MODEL ($input, vocab, d\_model, num\_heads, d\_ff, num\_blocks, dropout, m\_len$)
13:     $input\_ids \leftarrow$ Input($shape = input; name =$ "input_ids")
14:     $attention\_mask \leftarrow$ Input($shape = input; name =$ "attention_mask")
15:     $embedding\_layer \leftarrow$ Embedding($input\_dim = vocab, output\_dim = d\_model$)($input\_ids$)
16:     $x \leftarrow embedding\_layer$
17:     **for** $i \leftarrow 1$ to $num\_blocks$ **do**
18:         $xl\_block \leftarrow$ XLBlock($d\_model, num\_heads, d\_ff, dropout, m\_len$)
19:         $x \leftarrow xl\_block(x)$
20:     **end for**
21:     $x \leftarrow$ Dense($128$, activation $= 'relu'$)($x$)
22:     $x \leftarrow$ Reshape($(-1, 64)$)($x$)
23:     $x \leftarrow$ Bidirectional(GRU($64$, return_sequences $= True$))(x)
24:     $x \leftarrow$ Bidirectional(LSTM($64$, return_sequences $= True$))($x$)
25:     $x \leftarrow$ GlobalAveragePooling1D()($x$)
26:     $outputs \leftarrow$ Dense($5$, activation $= 'softmax'$)($x$)
27:     $model \leftarrow$ Model($inputs = [input\_ids, attention\_mask], outputs = outputs$)
28:     **return** $model$
29: **end function**
30: Initialize $input \leftarrow (120,)$
31: Initialize $vocab \leftarrow tokenizer.vocab\_size + 1$
32: $model \leftarrow$ model($input, vocab$)

## Datasets statistics

Table 1 presents a comprehensive summary of the Bengali and English datasets used in the research. The Bengali dataset has 44,001 texts with an average length of 75.51 words. The texts range from a minimum length of 0 words to a maximum length of 1,627 words. By contrast, the English dataset has 99,990 texts with an average length of 119.49 words. The largest length of a text is 763 words, while the shortest length is four words. Bengali has a larger standard deviation in text length (109.54) compared to English (77.14), suggesting a

**Table 1  Statistics of Bengali and English datasets.**

| Statistic | Bengali | English |
|---|---|---|
| Number of texts | 44,001 | 99,990 |
| Average length of text | 75.51 | 119.49 |
| Max length of text | 1,627 | 763 |
| Min length of text | 0 | 4 |
| Standard deviation of length of text | 109.54 | 77.14 |
| Median length of text | 40.0 | 96.0 |
| 25th percentile of length of text | 20.0 | 58.0 |
| 50th percentile of length of text | 40.0 | 96.0 |
| 75th percentile of length of text | 83.0 | 164.0 |
| 100th percentile of length of text | 1,627.0 | 763.0 |
| Number of unique words | 56,199 | 270,439 |
| Total number of words | 535,585 | 2,132,882 |

greater degree of variation in text lengths. In addition, the English dataset has a much greater number of distinct terms (270,439) and total words (2,132,882) in comparison to the Bengali dataset, which has 56,199 unique words and a total of 535,585 words. These statistics demonstrate the variations in the distribution of text length and the level of lexical diversity between the two datasets.

## Data preprocessing

### Text cleaning

The dataset used in this study was initially in a raw format and contained various issues such as punctuation, emojis, English characters, extra spaces, and single Bengali characters. To enhance its compatibility and efficiency for cyberbullying detection, a thorough cleaning process was conducted. Excessive punctuation can complicate the identification of offensive language, as seen in the Bengali phrase "ওই হালার পুত" (meaning 'the son of a bas***'), prompting the removal of symbols like "/::", "/::", "(-_-)", and others. Emojis, which convey emotions, were eliminated as they were irrelevant to the study. English characters were also removed to reduce ambiguity in reading Bengali comments, while extra spaces that could obscure meaning were addressed. For example, the comment "নাস্তিকের বাচ্চা নাস্তিক" (meaning 'sons of atheists') could appear as "নাস্তিকের বাচ্চা * * নাস্তিক" with excess spacing. Finally, single Bengali characters were discarded due to their lack of meaningful content. This cleaning process was essential to ensure the integrity and clarity of the dataset for cyberbullying detection in Bengali.

## Stop words removing

The lack of a standardized list of stop words makes it particularly difficult to recognize and deal with stop words in the Bengali language (*Mahmud, Ptaszynski & Masui, 2024*). We used a manual method to purify datasets for insightful analysis, meticulously compiling a list of stop words using in-depth regular expression analysis.

Our curated list includes words such as (*e.g.*, 'অজ', 'অজগর', 'অঞ্চ', 'অট', 'অটল', 'অড', এমব', 'এমর', 'এর', 'এরকম, 'হব', 'হস', 'হসক', 'হসকন) and more. These stop words are phrases that are often used but are judged to have negligible worth in a particular context. These were found after careful linguistic study. We had to rely on human analysis and create a custom stop word list that was suited to the peculiarities of Bengali because there was no pre-compiled list of stop words appropriate to the language. Consequently, a comprehensive refining procedure was applied to the dataset, removing all instances of these stop words. By doing this, we ensured that the dataset was free of irrelevant data, making it possible to examine the underlying data in a more meaningful and targeted manner. Our study is more accurate and pertinent because of this careful curation, which concentrates on important details within the framework of the Bengali language.

## Class imbalance mitigation through random oversampling

To address the class imbalance in our dataset, we employed random oversampling, which aims to balance class representation (*Saadi & Dhannoon, 2023*). Class imbalance in machine learning occurs when certain classes are underrepresented, leading to biased model performance (*Rafi-Ur-Rashid, Mahbub & Adnan, 2022*). We utilized the RandomOverSampler function from the imbalanced-learn package, which randomly selects examples from minority classes and duplicates them until class distribution is balanced. Mathematically, the procedure can be described as follows:

$$N_{\text{original}} = \sum_{i=1}^{C} n_i$$

$$N_{\text{target}} = \max(n_1, n_2, \ldots, n_C).$$

Here, $N_{\text{original}}$ represents the total instances in the dataset, $n_i$ denotes the instances in class $i$, and $C$ is the total number of classes. The oversampling process aims to increase the instances in minority classes to match the number in the majority class ($N_{\text{target}}$).

## Data augmentation

Synonym replacement is a key technique in Bengali text augmentation that enhances vocabulary while preserving semantic meaning. Given an original Bengali text $T_{\text{orig}}$ with $n$ words $w_i$, the goal is to substitute each word with a synonym if available:

$$T_{\text{aug}} = \begin{cases} s_i & \text{if synonyms}(w_i) \text{ exists} \\ w_i & \text{otherwise} \end{cases}_{i=1}^{n}.$$

Here, $s_i$ represents a randomly chosen synonym for the $i^{th}$ word $w_i$. Resources like Bengali WordNet (*Majumder et al., 2022*) are vital for identifying synonyms. If no synonyms exist, the original word is retained to maintain semantic coherence.

Given the linguistic diversity of Bengali, it is crucial to consider cultural and regional nuances during synonym substitution. For instance, the term "খুশি" (khushi), meaning "happy," can be replaced with synonyms like "আনন্দিত" (anondito) or "উৎসবগ্রস্ত" (utsobgrashto), each carrying distinct emotional connotations. Thus, synonym

replacement not only broadens vocabulary but also captures the subtleties of the Bengali language.

## Text tokenization

In this study, we developed tokenizers specifically for the Bengali language to handle a wide range of linguistic data efficiently. The BERT tokenizer was utilized to encode Bengali text entries into input IDs and attention masks. The tokenizer was pre-trained using Bengali language data (*Sarker, 2020*).

Following tokenization, each language dataset training, validation, and test dataset was given its distinct array, including the input IDs and attention masks. The arrays were transformed into NumPy arrays to make processing and manipulation efficient. Any extraneous dimensions were eliminated to optimize the input data format. The resultant tokenized input arrays were then prepared using BERT-based models. In this study, the dataset was divided into 70:15:15 ratios for training, testing, and validation, respectively.

## Base line models

### Transformer

The Transformer model plays a crucial role in detecting cases of cyberbullying in this study. It utilizes self-attention processes and feed-forward networks to record complex connections and contextual information in written language accurately. Layer normalization and dropout are key aspects that improve stability. Utilizing TensorFlow and Keras, this system operates on tokenized text sequences and attention masks, transforming input tokens into compact vectors. Global average pooling produces fixed-length representations that may be used for classification with softmax activation.

### Transformer-GRU

The Transformer-GRU model, which combines Transformer and GRU layers, is specifically designed to detect cyberbullying. The tokenized text sequences transform embeddings, which are then processed by Transformer blocks to extract features. Global average pooling generates fixed-size representations for classification using softmax activation. A Reshape layer is used to modify the output to enhance contextual awareness *via* the recurring nature of the GRU layer. This model is built using the Adam optimizer and categorical cross-entropy loss.

### Transformer-LSTM

In this work, the researchers have developed a model called the Transformer-LSTM, which combines the Transformer and LSTM layers. This model is specifically designed to detect instances of cyberbullying. First, tokenized text sequences are transformed into embeddings, which are further analyzed using Transformer blocks to extract features. Global average pooling produces fixed-length representations that are then used for classification using softmax activation. The output is reconfigured to align with the input specifications of the LSTM layer, which enhances contextual comprehension *via* its iterative nature.

### Transformer-GRU-LSTM

The Hybrid Transformer model, designed for cyberbullying detection, integrates Transformer, GRU, and LSTM layers. The process begins with embedding tokenized text sequences, which are processed through multiple Transformer blocks to capture complex relationships and contextual information. Global average pooling is then applied to obtain fixed-length feature representations. These features flow through dense layers before being reshaped for the GRU layer, which enhances the understanding of temporal patterns. An LSTM layer further improves sequence comprehension. The final representation is classified using a softmax activation function. Compiled with the Adam optimizer and categorical cross-entropy loss, this hybrid architecture aims for robust and accurate detection of cyberbullying in Bengali literature.

### Transformer-BiGRU-BiLSTM

Transformer blocks with bidirectional GRU and LSTM layers are integrated in the Hybrid Transformer model. To create fixed-length feature representations, tokenized text sequences are first transformed through an embedding layer and then global average pooling is used. The data is reshaped for additional processing by a thick layer. The comprehension of temporal patterns from both directions is improved with the addition of bidirectional LSTM and GRU layers. Ultimately, a softmax layer receives the combined representation to classify it.

### Transformer-XL

Transformer-XL uses sophisticated blocks with relative positional encoding. A tokenized text sequence embedding layer is the first step, and then other Transformer-XL blocks are added to handle long-range dependencies and contextual data. These blocks include feed-forward networks, multi-head self-attention, and layer normalization. Word locations in sequences can be more accurately identified by the model thanks to the relative positional encoding. Fixe-length feature representations are produced by a global average pooling operation following processing through the Transformer-XL blocks. A softmax layer is then used to classify the results. Cyberbullying in Bengali literature may be reliably identified with this model thanks to its compilation using the Adam optimizer and categorical cross-entropy loss.

## Hyperperameter settings

Table 2 summarizes our hyperparameter configurations for the models used in this research. Each row represents a distinct hyperparameter, such as embedding size, attention heads, feed-forward dimensions, and other related parameters, while the columns list the models, including Transformer, Transformer-GRU, Transformer-LSTM, Transformer-GRU-LSTM, Transformer-BiGRU-BiLSTM, Transformer-XL, and Fusion Transformer-XL. We select these hyperparameters to optimize performance while maintaining computational efficiency. The baseline models use an embedding dimension of 128, whereas we set Fusion Transformer-XL to 256 to enhance contextual representation in Bengali, which exhibits greater morphological complexity than English. We maintain two

**Table 2 Detailed hyperparameter settings for various models used in the study.** (The hyperparameters include the T = Transformer, M = Millions).

|  | T | T-GRU | T-LSTM | T-GRU-LSTM | T-BiGRU-BiLSTM | T-XL | Fusion T-XL |
|---|---|---|---|---|---|---|---|
| Embedding dimension | 128 | 128 | 128 | 128 | 128 | 128 | 256 |
| Number of heads | 2 | 4 | 4 | 4 | 4 | 4 | 2 |
| Feed-forward dimension | 128 | 128 | 128 | 128 | 128 | 128 | 256 |
| Number of transformer blocks | 2 | 2 | 2 | 2 | 2 | 2 | 2 |
| Dropout rate | 0.1 | 0.1 | 0.1 | 0.1 | 0.1 | 0.1 | 0.2 |
| GRU units | – | 64 | – | 64 | 64 (BiGRU) | – | 64 (BiGRU) |
| LSTM units | – | – | 64 | 64 | 64 (BiLSTM) | – | 64 (BiLSTM) |
| Memory length | – | – | – | – | – | 128 | 128 |
| Relative positional encoding | No | No | No | No | No | Yes | Yes |
| Optimizer | Adam | Adam | Adam | Adam | Nadam | Adam | Adam |
| Batch sizes | 32 | 32 | 32 | 32 | 32 | 32 | 64 |
| Trainable parameters | 13.39 M | 13.68 M | 13.69 M | 13.71 M | 13.78 M | 13.64 M | 28.66 M |

transformer blocks across all models to prevent overfitting while preserving adequate representational capacity. We adjust the multi-head attention mechanism to control training parallelization, setting Fusion Transformer-XL to two heads to ensure stable gradient propagation and reduce computational complexity. We set the feed-forward dimension to 256 in Fusion Transformer-XL to improve feature representation without excessive parameter expansion. Specific models incorporate BiGRU and BiLSTM layers to enhance contextual understanding, with 64 units selected for efficient sequence learning while maintaining computational feasibility. We utilize relative positional encoding in Transformer-XL and Fusion Transformer-XL to capture long-range dependencies, which are crucial for cyberbullying detection. We apply a dropout rate of 0.1 for most models to mitigate overfitting, while Fusion Transformer-XL uses 0.2 to regularize its larger parameter set. We set the memory length to 128 to ensure effective context retention. We use the Adam optimizer for all models except Transformer-BiGRU-BiLSTM, which employs Nadam for better adaptive learning rate adjustments. Fusion Transformer-XL adopts a batch size of 64 to accommodate its increased complexity while maintaining training stability. With 28.66 million trainable parameters, Fusion Transformer-XL significantly exceeds the baseline models, reflecting its enhanced capacity to capture cyberbullying patterns. These choices strike a balance between computational efficiency and performance, making our model suitable for Bengali and other low-resource languages. Future research should explore hyperparameter tuning and alternative architectures.

## Training setup

In this section, we present a comprehensive assessment of our proposed Fusion Transformer-XL model for detecting cyberbullying in the Bengali language. We conduct the evaluation using computing resources such as an AMD Ryzen CPU, 16GB of RAM

operating at a clock speed of 3,200 Hz, and an RTX GeForce 2060 GPU with 12GB of memory. We use Python version 3.11.9 and TensorFlow version 2.15.0 to refine the model's parameters using methods such as Adam optimisation and early stopping regularisation. During the training process, we closely track important metrics such as accuracy, precision, recall, and F1 score to evaluate the effectiveness of the model. Additionally, we use k-fold cross-validation to ensure the model's reliability and stability. By looking at the results of the experiment, we look into how hyperparameters like the number of transformer-XL blocks and attention processes affect the results. This will help us understand model's efficiency and applicability in real-world settings for detecting cyberbullying in Bengali text data.

## Evaluation metrics

The focus is on evaluating the efficacy of our suggested hybrid transformer-XL model *via* the use of several performance indicators. These measures, such as accuracy ($Acc$), precision ($P$), recall ($R$), and F1-score ($F1$), are crucial indications of the model's prediction skills and its capacity to identify instances of cyberbullying in Bengali text data. Mathematically, accuracy is quantitatively determined by dividing the number of properly predicted cases by the total number of instances.

$$Acc = \frac{TP + TN}{TP + TN + FP + FN}.$$

The variables $TP$, $TN$, $FP$, and $FN$ represent true positives, true negatives, false positives, and false negatives, respectively. Precision is a measure that calculates the ratio of correct positive predictions to all positive predictions:

$$P = \frac{TP}{TP + FP}.$$

While recall measures the proportion of true positive predictions among all actual positive instances:

$$R = \frac{TP}{TP + FN}.$$

The F1-score, the harmonic mean of precision and recall, provides a balanced assessment of the model's performance:

$$F1 = 2\frac{PR}{P + R}.$$

The performance study highlights the hybrid transformer-XL model's capacity to handle the complexity of Bengali cyberbullying detection. The model obtains a high F1-Score by skilfully balancing recall and accuracy, showing that it can reliably detect actual cases of cyberbullying in addition to making correct predictions. This thorough assessment shows that the model has the potential to be a dependable instrument for automated detection and intervention, which will help create safer online settings.

**Table 3** **This table presents a comparison of the performance of several models on datasets in Bengali and English, both with and without the use of data augmentation methods.** The findings include accuracy measures for the training, validation, and test stages. This comprehensive comparison facilitates the comprehension of the influence of data augmentation on several models across two separate datasets. The bolded accuracy values in the table highlight the highest test accuracies achieved among the models on each dataset.

| Model name | Aug | Bengali dataset | | | English dataset | | |
|---|---|---|---|---|---|---|---|
| | | Train Acc | Val Acc | Test Acc | Train Acc | Val Acc | Test Acc |
| T | 0 | 92.74% | 88.39% | 87.56% | 93.76% | 93.30% | 93.14% |
| | 1 | 98.11% | 97.33% | 97.21% | 98.64% | 98.93% | 98.66% |
| T-GRU | 0 | 94.55% | 88.77% | 88.34% | 97.86% | 98.46% | 98.31% |
| | 1 | 98.04% | 97.36% | 97.20% | 98.75% | 98.81% | 98.65% |
| T-LSTM | 0 | 94.31% | 88.59% | 88.48% | 97.05% | 98.05% | 97.77% |
| | 1 | 97.44% | 96.63% | 96.63% | 98.64% | 98.59% | 98.61% |
| T-GRU-LSTM | 0 | 84.64% | 80.90% | 82.46% | 93.72% | 82.37% | 82.09% |
| | 1 | 96.26% | 95.60% | 95.42% | 97.86% | 97.98% | 97.93% |
| TGL | 0 | 94.22% | 88.25% | 88.97% | 97.78% | 98.08% | 97.67% |
| | 1 | 98.08% | 97.54% | 97.46% | 98.14% | 98.38% | 98.36% |
| T-XL | 0 | 93.68% | 88.90% | 88.34% | 97.25% | 96.44% | 96.54% |
| | 1 | 97.94% | 97.62% | 97.46% | 91.37% | 92.66% | 92.58% |
| FTXL | 0 | 93.92% | 88.34% | 88.36% | 98.78% | 98.30% | 98.37% |
| | 1 | **98.63%** | **98.16%** | **98.17%** | **99.85%** | **99.60%** | **99.65%** |

**Note:**
T, Transformer; FTXL, Fusion Transformer-XL; TGL, Transformer-BiGRU-BiLSTM; Aug, Augmentation; Acc, Accuracy; Val, Validation.

# RESULT ANALYSIS

## Comparison with baseline models

The performance of several neural network models on Bengali and English datasets is compared in Table 3, both with and without data augmentation. The available models are Transformer, Transformer-GRU, Transformer-LSTM, Transformer-GRU-LSTM, Transformer-BiGRU-BiLSTM, Transformer-XL, and Fusion Transformer-XL. Each model's accuracy is tracked throughout training, validation, and testing. Data augmentation, as shown in the table provided, greatly enhances performance for all models. The bolded accuracy values in the table highlight the highest test accuracies achieved among the models on each dataset. Notably, the Fusion Transformer-XL model demonstrates superior performance, attaining a test accuracy of 98.13% and 99.65% on the Bengali and English datasets, respectively, when augmentation is used. In comparison, without augmentation, the model achieves accuracies of 88.36% and 99.00% for the same datasets. This demonstrates the efficacy of data augmentation in improving the overall performance of models and identifies the Fusion Transformer-XL as the most resilient model architecture for natural language processing tasks.

Table 4 presents a performance comparison of several models on Bengali and English datasets, with and without data augmentation. The models available include Transformer (T), Transformer-GRU (T-GRU), Transformer-LSTM (T-LSTM), Transformer-GRU-

**Table 4 This table displays a comparative analysis of the efficacy of several models on datasets in Bengali and English, both with and without the use of data augmentation techniques.** The measurements consist of precision (P), recall (R), and F1-score (F1) for the training, validation, and test phases. This extensive comparison enhances understanding of the impact of data augmentation on many models across two distinct datasets. The bolded precision, recall, and F1-score values in the table indicate the highest scores achieved among the models.

| Models | Aug | Bengali dataset | | | English dataset | | |
|---|---|---|---|---|---|---|---|
| | | P | R | F1 | P | R | F1 |
| T | 0 | 87.40% | 87.55% | 87.41% | 93.60% | 92.29% | 93.75% |
| | 1 | 97.25% | 97.25% | 97.25% | 98.52% | 98.43% | 98.47% |
| T-GRU | 0 | 88.36% | 88.35% | 88.22% | 97.68% | 97.30% | 97.48% |
| | 1 | 97.22% | 97.20% | 97.20% | 98.88% | 98.58% | 98.73% |
| T-LSTM | 0 | 88.46% | 88.47% | 88.40% | 97.44% | 97.69% | 97.56% |
| | 1 | 96.65% | 96.64% | 96.62% | 98.28% | 98.66% | 98.47% |
| T-GL | 0 | 82.90% | 82.45% | 82.48% | 97.24% | 97.16% | 97.19% |
| | 1 | 95.42% | 95.43% | 95.42% | 97.70% | 97.99% | 97.92% |
| TGL | 0 | 88.94% | 88.96% | 88.95% | 97.48% | 97.89% | 97.68% |
| | 1 | 97.45% | 97.46% | 97.46% | 98.14% | 98.42% | 98.28% |
| T-XL | 0 | 88.22% | 88.33% | 88.21% | 96.82% | 97.39% | 97.09% |
| | 1 | 97.46% | 97.45% | 97.46% | 95.03% | 89.03% | 91.47% |
| FTXL | 0 | 88.27% | 88.36% | 88.26% | 98.87% | 98.22% | 98.36% |
| | 1 | **98.18%** | **98.18%** | **98.18%** | **99.56%** | **99.62%** | **99.59%** |

**Note:**
Aug, Augmentation; Without Augmentation = 0, With Augmentation = 1, T, Transformer; T-GL, Transformer-GRU-LSTM; TGL, Transformer-BiGRU-BiLSTM.

LSTM (T-GRU-LSTM), Transformer-BiGRU-BiLSTM (TGL), Transformer-XL (T-XL), and Fusion Transformer-XL (FTXL). Performance is evaluated using precision (P), recall (R), and F1-score (F1). Data augmentation greatly improves all metrics. The Fusion Transformer-XL model with augmentation (FTXL, Aug = 1) obtained the greatest scores: 98.13% precision (P), 98.11% recall (R), and 98.12% F1 score for Bengali; and 99.56% precision (P), 99.62% recall (R), and 99.59% F1 score for English. This underscores the efficacy of data augmentation in enhancing model precision and resilience.

## Proposed fusion transformer-XL performance metrics

Figure 2 displays four confusion matrices that demonstrate the effectiveness of a fusion transformer model in recognising instances of cyberbullying in both Bengali and English languages. Figures 2A and 2B illustrate the model's performance in Bengali, with (a) using data augmentation and (b) without it. In scenario (a), the model accurately categorises incidents as "Not Bully" 4,630 times, "Religious" 4,447 times, "Sexual" 4,468 times, "Threat" 4,410 times, and "Troll" 4,421 times. In scenario (b), where data augmentation is not used, the number of accurate classifications decreases to 1,397 for the category "Not Bully," 1,465 for "Religious," 1,186 for "Sexual," 1,509 for "Threat," and 1,214 for "Troll." Figures 2C and 2D illustrate the model's performance in English, comparing situations with and without data augmentation. In condition (c), the model attains accurate

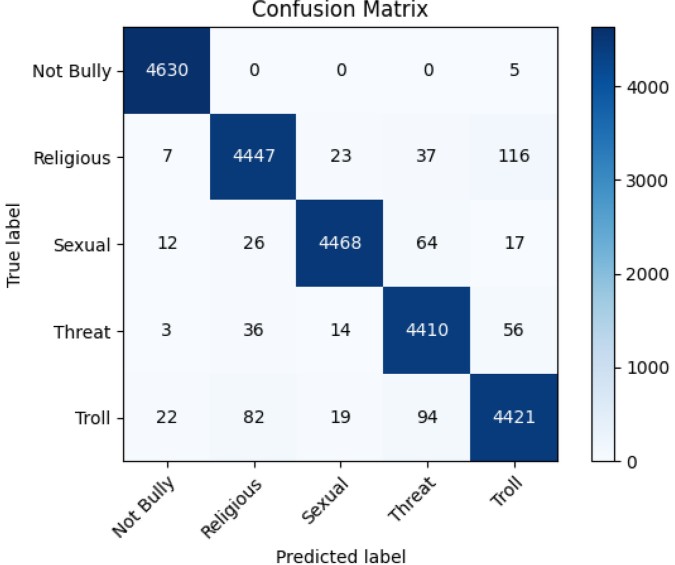

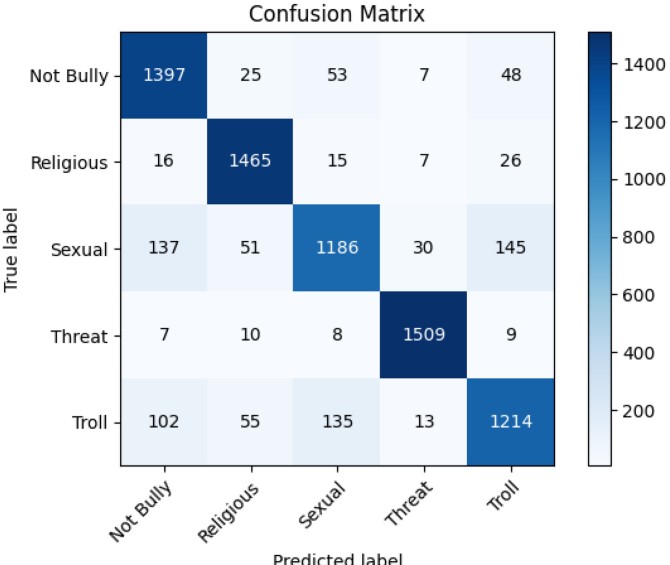

**(a)** The confusion matrix demonstrates the effectiveness of the fusion transformer model in detecting Bengali cyberbullying, taking into account the use of data augmentation.

**(b)** Confusion matrix showing the effectiveness of the fusion transformer model for the identification of Bengali cyberbullying in the absence of data augmentation

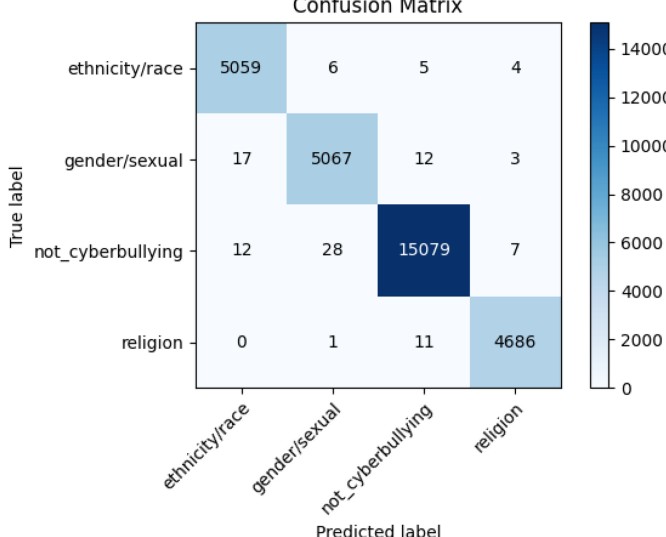

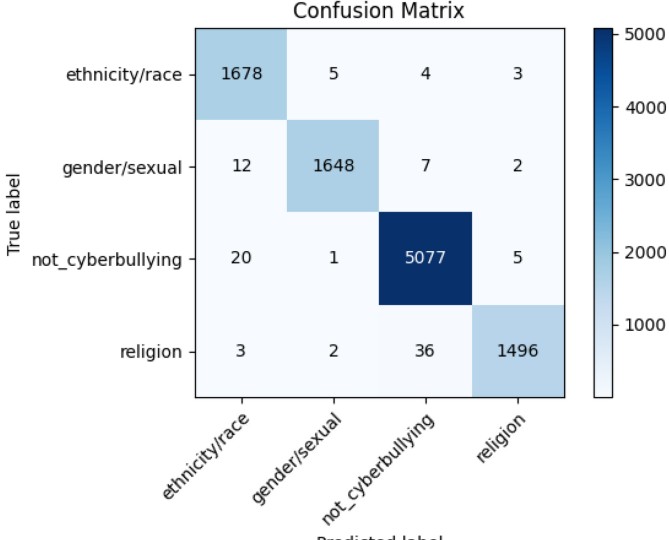

**(c)** The confusion matrix demonstrates the effectiveness of the fusion transformer-xl model in detecting English cyberbullying, taking into account the use of data augmentation.

**(d)** Confusion matrix showing the effectiveness of the fusion transformer-xl model for the identification of English cyberbullying in the absence of data augmentation

**Figure 2** The diagram depicts the distribution of actual *vs.* anticipated labels across several categories, providing a visual representation of the model's classification precision and the nature of the mistakes made.

categorizations of 5,059 instances for "ethnicity/race," 5,067 instances for "gender/sexual," 15,079 instances for "not_cyberbullying," and 4,686 instances for "religion." In the absence of data augmentation in (d), the number of accurate categories decreases to 1,678 for

"ethnicity/race," 1,648 for "gender/sexual," 5,077 for "not_cyberbullying," and 1,496 for "religion." The findings suggest that the use of data augmentation significantly improves the accuracy of the model's classification in both languages, as shown by the higher number of right predictions (diagonal values) and lower number of misclassifications (off-diagonal values). The visual representations emphasize the vital importance of data augmentation in enhancing the efficacy of cyberbullying detection methods.

Figure 3 exhibits four receiver operating characteristic (ROC) curves that assess the efficacy of the fusion transformer model in identifying cyberbullying in both Bengali and English scenarios. Figure 3A demonstrates the efficacy of the model in detecting Bengali cyberbullying with data augmentation. It achieves flawless classification with an area under the curve (AUC) of 1.00 for all categories, including "Not Bully," "Troll," "Sexual," "Religious," and "Threat." Figure 3B displays the model's performance when data augmentation is not used. The AUC values for "Not Bully," "Troll," "Sexual," "Religious," and "Threat" are marginally decreased to 0.98, 0.97, 0.95, 0.99, and 1.00, respectively. The performance of the model in English cyberbullying detection with data augmentation is shown in Fig. 3C. It achieves an AUC of 1.00 across all classes, including "ethnicity/race," "gender/sexual," "not_cyberbullying," and "religion." Figure 3D displays the model's performance when data augmentation is not used. It maintains an AUC of 1.00 for most classes but slightly decreases to 0.99 for the "religion" class. The ROC curves showcase the exceptional efficacy of the fusion transformer model, especially when combined with data augmentation. This demonstrates its strong capacity to reliably detect different types of cyberbullying in several languages.

## Baseline models performance metrics

The confusion matrices in Fig. 4 illustrate how well different transformer-based models perform when it comes to identifying Bengali cyberbullying with data augmentation. A to F, the sub-figures, each represent a distinct model: Transformer in Fig. 4A, Transformer-GRU in Fig. 4B, Transformer-LSTM in Fig. 4C, Transformer-GRU-LSTM in Fig. 4D, Transformer-BiGRU-BiLSTM in Fig. 4E, and Transformer-XL in Fig. 4F. The distribution of genuine labels against predicted labels is depicted in the diagrams, giving a clear picture of each model's accuracy and the many kinds of mistakes that might occur. This emphasizes how well data augmentation can enhance model performance.

With the use of data augmentation, Fig. 5 displays a number of confusion matrices that show how well different transformer-based models do in identifying English cyberbullying across a variety of categories, including religion, non-cyberbullying, gender/sexual orientation, and ethnicity/race. The results for the transformer model are shown in Fig. 5A, the transformer-GRU model in Fig. 5B, the transformer-LSTM model in Fig. 5C, the transformer-GRU-LSTM model in Fig. 5D, the transformer-BGRU-BiLSTM model in Fig. 5E, and the transformer-XL model in Fig. 5F. By comparing the real and predicted labels in each confusion matrix, the model's precision in classifying cyberbullying cases with data augmentation is brought to light. The matrices indicate the number of correctly classified cases along the diagonal and the number of misclassified cases off-diagonal, shedding light on the relative efficacy of each model and the regions with higher rates of

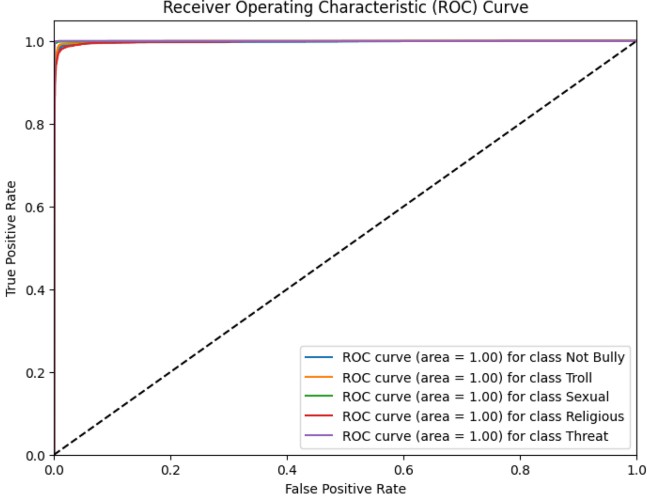
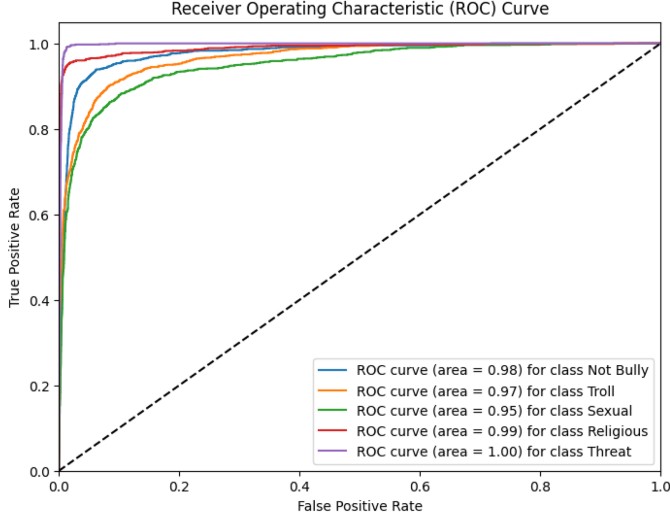

(a) The ROC curve demonstrates the effectiveness of the fusion transformer-xl model in detecting Bengali cyberbullying, taking into account the use of data augmentation

(b) ROC curve showing the effectiveness of the fusion transformer-xl model for the identification of Bengali cyberbullying in the absence of data augmentation.

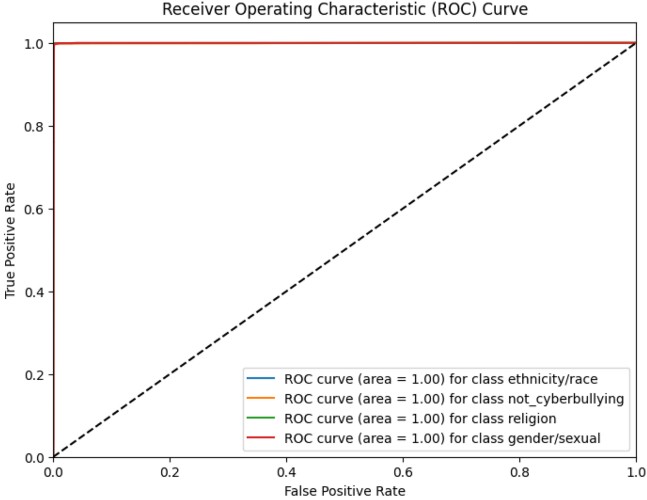
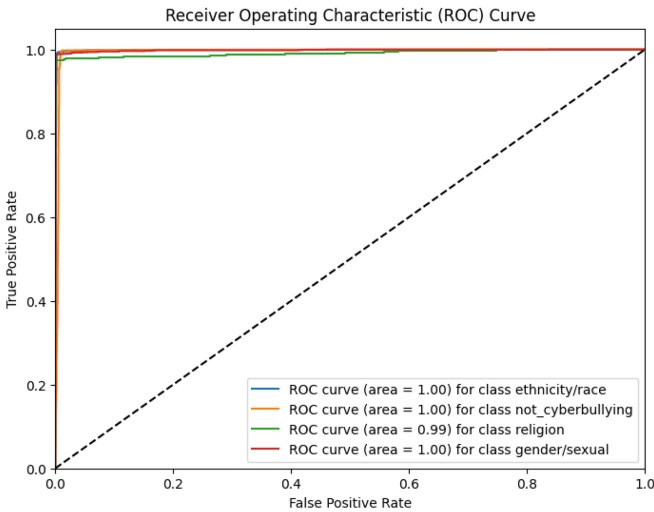

(c) The ROC curve demonstrates the effectiveness of the fusion transformer-xl model in detecting English cyberbullying, taking into account the use of data augmentation.

(d) ROC curve showing the effectiveness of the fusion transformer-xl model for the identification of English cyberbullying in the absence of data augmentation.

**Figure 3** **ROC curves were generated for the Fusion Transformer-XL model, representing its performance across different classes.** Each curve corresponds to the model's classification performance.

misclassification. Together, these visualisations highlight how different algorithms perform in terms of accurately recognising cyberbullying incidents in the enriched dataset.

A set of receiver operating characteristic (ROC) curves is shown in Fig. 6, which illustrates how well different transformer-based models identify Bengali cyberbullying in several classes when data is supplemented. The ROC curve for the transformer model is specifically shown in Figs. 6A–6F for the transformer, transformer-GRU, LSTM,

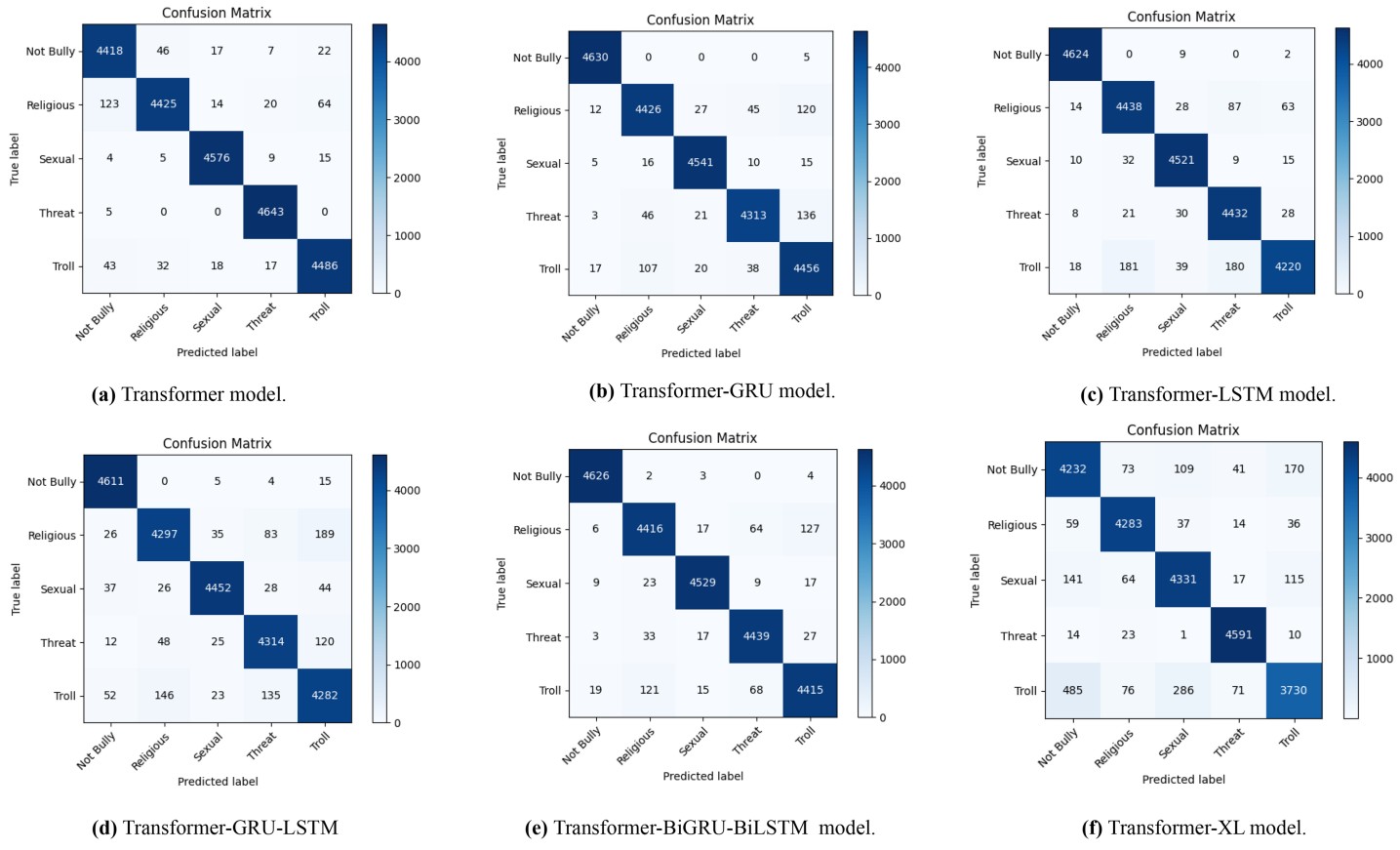

**(a)** Transformer model.

**(b)** Transformer-GRU model.

**(c)** Transformer-LSTM model.

**(d)** Transformer-GRU-LSTM

**(e)** Transformer-BiGRU-BiLSTM model.

**(f)** Transformer-XL model.

**Figure 4** The diagram shows the distribution of actual *vs.* predicted labels, illustrating the baseline model's classification accuracy and error patterns in Bengali with data augmentation.

transformer-GRU-LSTM, BGRU-BiLSTM and Transformer-XL models, respectively. Plotting the true positive rate *vs.* the false positive rate for each class—including threat, gender/sexual orientation, ethnicity/race, and not cyberbullying—is done using a ROC curve. The models' performance is shown by the AUC values in the legend; greater AUC values correspond to better performance. When data augmentation is performed, the curves show that all models have high AUC values across the classes, demonstrating a significant capacity to discern between the various forms of cyberbullying. The combined impact of these visualisations is to show how different models differ in how well they can detect Bengali cyberbullying cases in the expanded dataset.

With data augmentation, Fig. 7 displays a series of ROC curves showing how well different transformer-based models identify English cyberbullying in various classes. The ROC curve for the transformer model is specifically shown in Figs. 7A–7F, for the transformer, transformer-GRU, LSTM, transformer-GRU-LSTM, BGRU-BiLSTM and Transformer-XL models, respectively. Plotting the true positive rate *vs.* the false positive rate for each class-ethnicity/race, non-cyberbullying, religion, and gender/sexual orientation-is what each ROC curve does. The models' performance is shown by the AUC values in the legend; greater AUC values correspond to better performance.

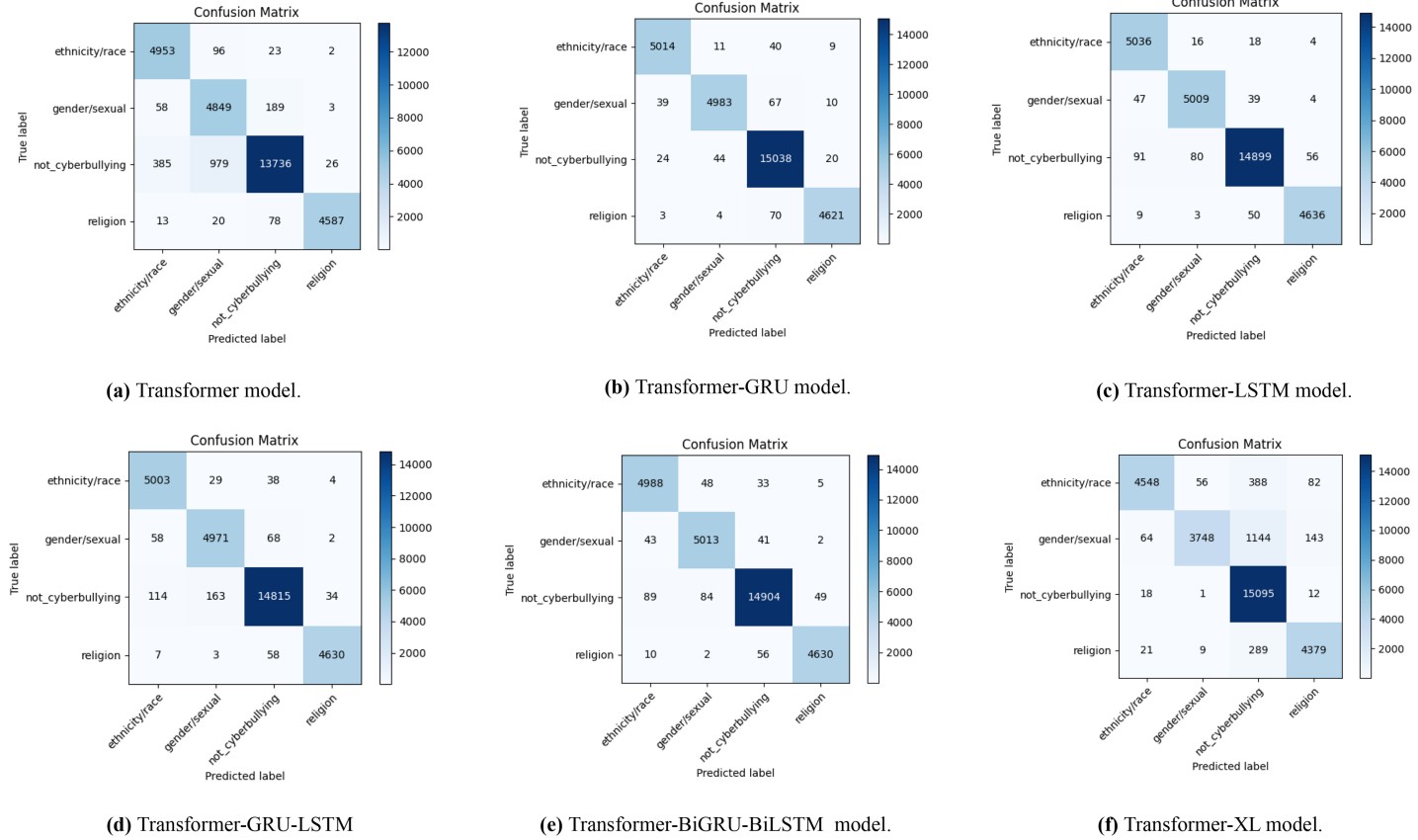

**(a)** Transformer model.  **(b)** Transformer-GRU model.  **(c)** Transformer-LSTM model.

**(d)** Transformer-GRU-LSTM  **(e)** Transformer-BiGRU-BiLSTM  model.  **(f)** Transformer-XL model.

**Figure 5** **The diagram shows the distribution of actual *vs.* predicted labels, illustrating the baseline model's classification accuracy and error patterns in English cyberbullying with data augmentation.**

According to the study, transformer-based models perform noticeably better at identifying cyberbullying in both Bengali and English datasets when data augmentation is used. Across categories including Not Bully, Religious, Gender/Sexual content, and Ethnicity/Race, precision, recall, and F1-score, as well as confusion matrices and ROC curves, demonstrate notable increases in model accuracy and dependability. Accurately identifying objectionable content is typically a greater ability of models trained with enriched data, as seen by their higher results. These findings demonstrate the importance of data augmentation in creating cyberbullying detection systems.

## K-fold cross-valildation

Table 5 demonstrates the influence of different numbers of folds in k-fold cross-validation on the metrics of model performance. This analysis offers valuable information on how various fold configurations impact the accuracy of training and validation, as well as the metrics of precision, recall, and F1 score. As the number of folds grows from 1 to 5, there is a little variation noticed in the training and validation accuracy rates. The greatest accuracy reached is 97.88% for training and 96.82% for validation, both occurring with two folds. Moreover, the accuracy, recall, and F1-score metrics consistently maintain their values
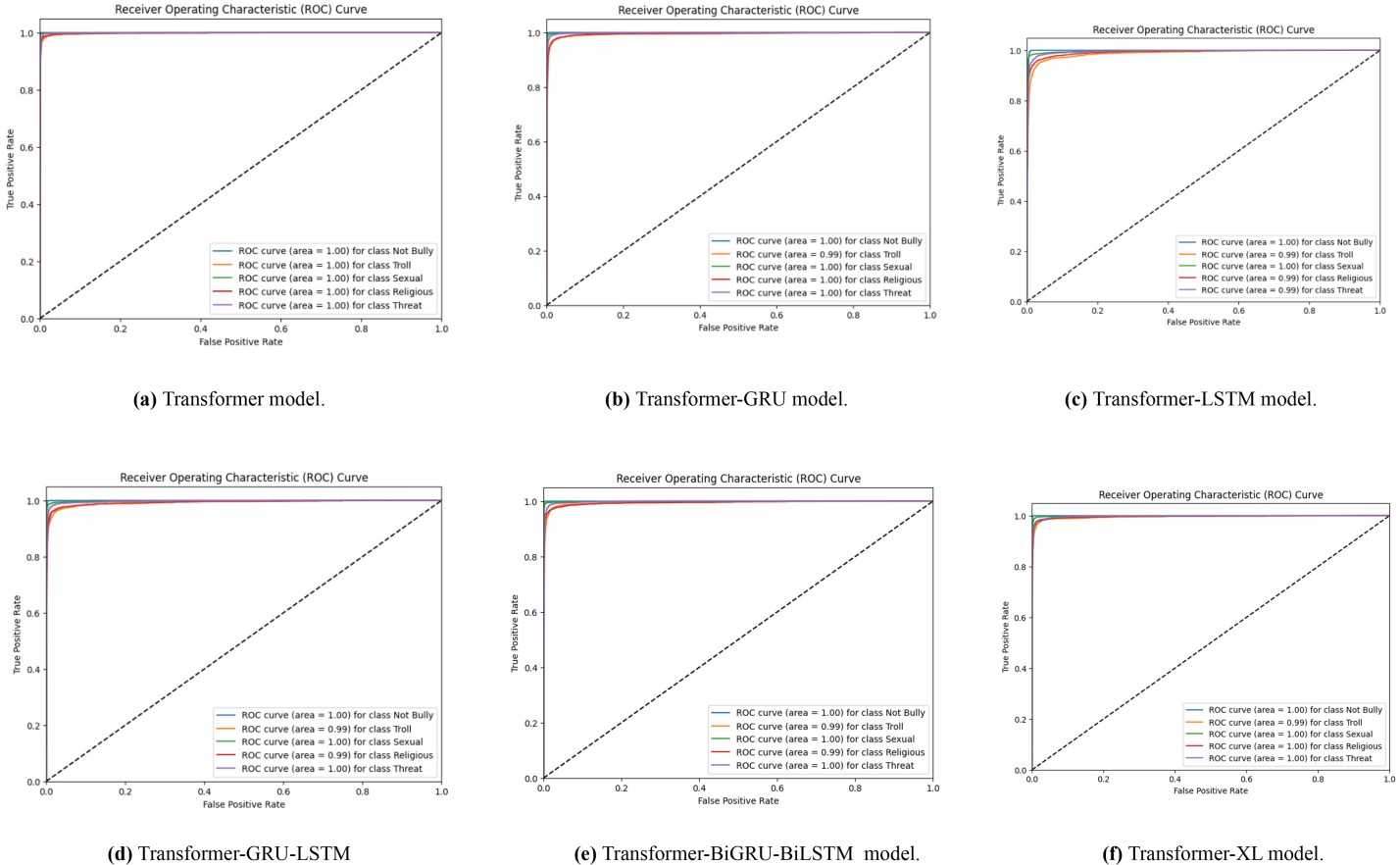

**(a)** Transformer model.

**(b)** Transformer-GRU model.

**(c)** Transformer-LSTM model.

**(d)** Transformer-GRU-LSTM

**(e)** Transformer-BiGRU-BiLSTM model.

**(f)** Transformer-XL model.

**Figure 6** **(A–F) ROC curves were generated for the baseline models, representing their performance across different classes.** Each curve corresponds to the model's classification performance in Bengali cyberbullying detection with data augmentation.

regardless of the fold settings, demonstrating the model's consistency in performance when subjected to multiple cross-validation procedures. In summary, this approach helps in choosing the best fold configuration for k-fold cross-validation, which guarantees a reliable assessment of the model and an accurate estimate of its performance.

## Analysis of model predictions on new text samples

Figure 8 shows both Bengali and English phrases were analyzed in the most current examination of the Fusion Transformer-XL model's predictions on unseen text samples to comprehend the model's decision-making process. For example, in the Bengali sentence Fig. 8A "আরেকটা বাংরেজি পণ্ডিত বের হইসে পুরাই ফাউল," "Another Bangree scholar came out full foul," key terms like "আরেকটা," "বাংরেজি" and "পণ্ডিত" played a crucial role in the text's categorization by the model. Similarly, the sentence "মেয়ে এক ইন্টারভিউতে বলেছে চোখে বিশ্বাস করেনা অর্থাৎ নাস্তিক অবিশ্বাসের ধর্ম ইসলাম ধর্মের রোজা রাখে এটাকি একটা", "The girl said in an interview that she does not believe in the eyes, that is, the religion of atheistic disbelief fasts in the religion of Islam" featured critical words like "মেয়ে", "নাস্তিক," and "ইসলাম," impacting the prediction outcome.

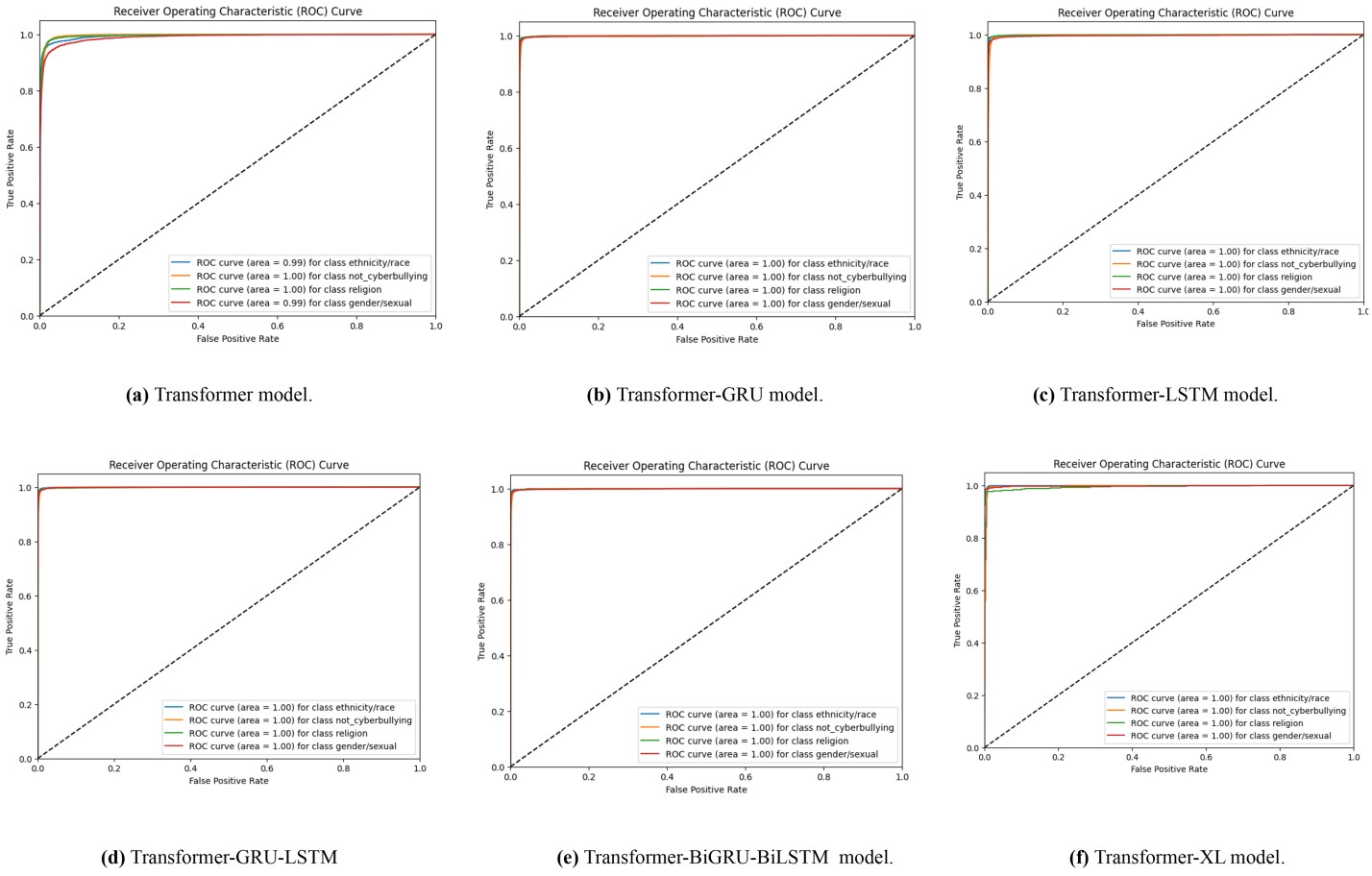

**(a)** Transformer model.

**(b)** Transformer-GRU model.

**(c)** Transformer-LSTM model.

**(d)** Transformer-GRU-LSTM

**(e)** Transformer-BiGRU-BiLSTM  model.

**(f)** Transformer-XL model.

**Figure 7** **(A–F) ROC curves were generated for the baseline models, representing its performance across different classes.** Each curve corresponds to the model's classification performance in English cyberbullying detection with data augmentation.

**Table 5 Impact of varying the number of folds in k-fold cross-validation on model performance metrics.**

| N-folds | Train acc | Val acc | P | R | F1 |
|---------|-----------|---------|------|------|------|
| 1 | 97.75% | 96.57% | 96.59% | 96.57% | 96.58% |
| 2 | 97.88% | 96.82% | 96.81% | 96.82% | 96.80% |
| 3 | 97.40% | 96.41% | 96.40% | 96.41% | 96.40% |
| 4 | 97.50% | 96.17% | 96.29% | 96.17% | 96.19% |
| 5 | 97.38% | 96.52% | 96.51% | 96.52% | 96.51% |

**Note:**
N-Folds, Number of folds; Acc, Accuracy; Val, Validation; P, Precision; R, Recall; F1, F1-Score.

In the English text depicted in Fig. 8B, words such as "zubear," "real," as well as "love," "hurt," and "asking," were highlighted from phrases like "all love does is get you hurt and leave you asking questions" and "zubear says any real isn't letting this happen." These terms were crucial in determining how the model classified the text.

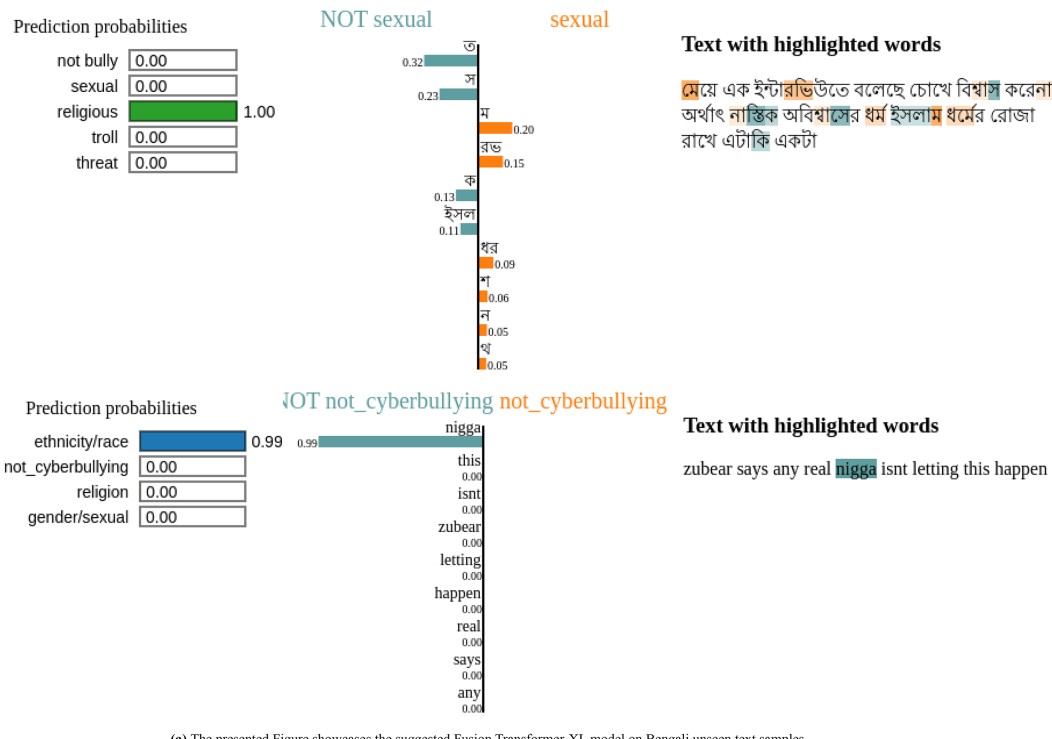

(a) The presented Figure showcases the suggested Fusion Transformer-XL model on Bengali unseen text samples.

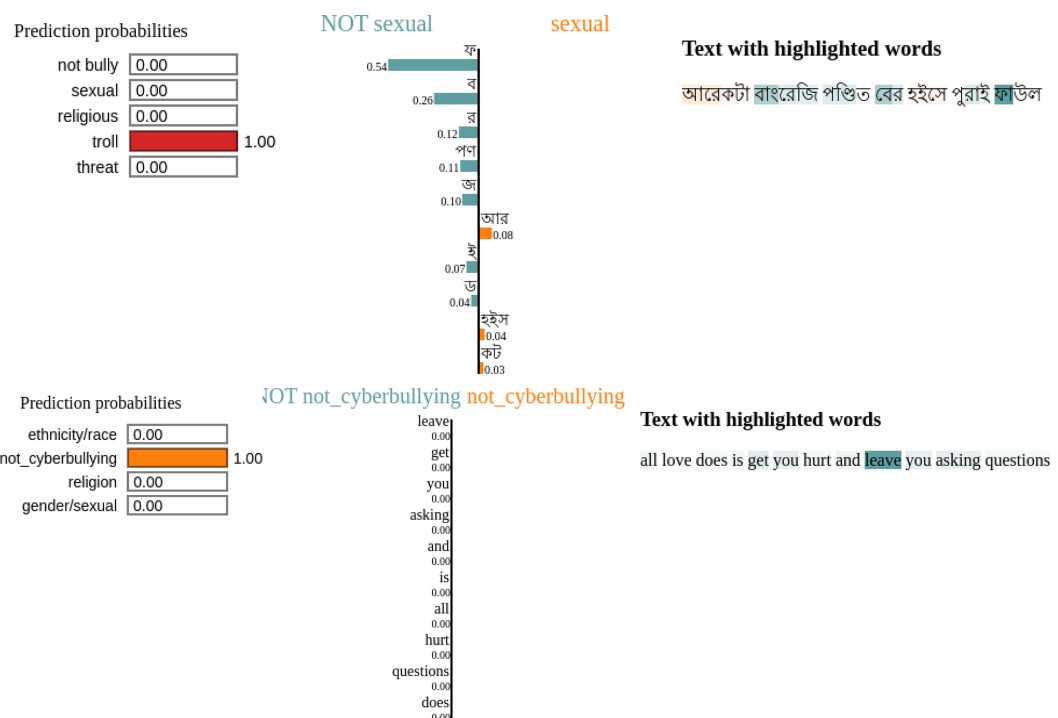

(b) The presented Figure showcases the suggested Fusion Transformer-XL model on English unseen text samples.

**Figure 8** **Fusion Transformer-XL model explanations for different Bengali and English text samples using LIME.**

**Table 6 Comparative analysis among existing work.**

| Reference | Applied classifier | Best classifier | Accuracy |
|---|---|---|---|
| *Ranasinghe & Zampieri (2020)* | XLM-R (TL), BERT-m (TL), XLM-R, BERT-m | BERT and XLM-R | Bengali: 84%, Hindi: 85% |
| *Kumar & Sachdeva (2022b)* | ConvNet, CapsNet | ConvNet, CapsNet | AUC-ROC of 0.98 |
| *Roy & Mali (2022)* | ConvNet, Transfer learning | Transfer Learning | 89% |
| *Teng & Varathan (2023)* | LR, Linear SVC, DistilBert, Electra-small, DistilRoBerta | DistilBert | 97.41% |
| *Ghosh et al. (2023)* | Attention-BiLSTM-CNN, BiLSTM-CNN, Attention-LSTM-CNN, LSTM-CNN, Attention-BiLSTM, BiLSTM, Attention-LSTM, LSTM | Attention-BiLSTM-CNN | 94.3% |
| *Nath, Karim & Miraz (2024)* | Bi-LSTM with momentum SGD, Bi-LSTM with Adam Optimiser, Bi-LSTM with 5-Fold Cross Validation | 2-layer Bi-LSTM | 95.08% |
| *Sihab-Us-Sakib et al. (2024)* | RF, SVM, MNB, GRU, CNN, LSTM, BiLSTM, m-BERT, Bangla-BERT, XLM-RoBERTa | XLM-RoBERTa | 82.61% |
| *Hoque & Seddiqui (2024)* | SVM, MNB, RF, LR, LSTM, BiLSTM, CNN-BiLSTM, BERT | BERT | 80.165% |
| *Diaz-Garcia & Carvalho (2025)* | SVM, Logistic regression, Naïve Bayes, CNNs, LSTMs, BiLSTMs, BERT, RoBERTa, GPT-2 | BERT | 97.34% |
| Our work | Transformer, Transformer-GRU, Transformer-LSTM, Transformer-GRU-LSTM, Transformer-BiGRU-BiLSTM, Transformer-XL, Fusion Transformer-XL | Fusion Transformer-XL | 98.17% |

To sum up, this method of using LIME explanations makes it clear which particular words and phrases have the most effect on the model's predictions. This approach facilitates the thorough evaluation of textual data for a range of categories, including trolling behaviour, religious context, and sexual content. Text categorization methods are more accurate and equitable when the influential terms in the sentences are highlighted. This makes it simpler to understand how the machine makes decisions.

## Comparative analysis among and existing works

Table 6 presents our comparative study, where we analyze various classifiers employed for cyberbullying detection across different research studies. The applied classifiers range from traditional machine learning models such as SVM, RF, and LR to advanced deep learning architectures, including ConvNet, CapsNet, and transformer-based models. Notably, *Teng & Varathan (2023)* achieved a high accuracy of 97.41% using DistilBert, while *Roy & Mali (2022)* and *Kumar & Sachdeva (2022b)* reported strong performance with Transfer Learning and CapsNet, respectively. Similarly, *Sihab-Us-Sakib et al. (2024)* employed XLM-RoBERTa and Bangla-BERT, achieving an accuracy of 82.61%, while *Diaz-Garcia & Carvalho (2025)* conducted a comparative analysis of cyberbullying detection methods using BERT, RoBERTa, and GPT-2, with BERT achieving 97.34% accuracy. Our study introduces a novel approach that leverages a fusion of Transformer-XL models, achieving an exceptional accuracy of 98.17%. This fusion strategy not only surpasses existing methods in accuracy but also enhances model robustness, contributing significantly to the advancement of cyberbullying detection techniques.

## DISCUSSION

In this section, we explore the implications of the results and the broader context of the research. The observed performance improvements with data augmentation highlight its effectiveness in addressing data scarcity concerns and improving the model's ability to handle textual material variances. Augmentation approaches enhance the model's ability to generalize to unseen cases by creating synthetic data instances that provide a more thorough representation of the underlying data distribution. Moreover, the variation in performance measures across various classes emphasizes the inherent difficulties in identifying cases of cyberbullying, especially when it comes to dealing with language-specific challenges in Bengali. Unlike English, Bengali presents morphological complexity, code-mixing (Bengali and English), and transliteration issues, which significantly impact model performance. Bengali words often have rich inflections and complex word structures, making tokenization and embedding generation more challenging. Additionally, users frequently incorporate English words or phrases into Bengali text, making it harder for the model to interpret semantic nuances accurately. The presence of Bengali text written in Romanized script (transliteration) further complicates detection, as the same word may appear in multiple forms (*e.g.*, "ভালবাসা" *vs.* "valobasha" for "love"), requiring enhanced preprocessing techniques.

The scarcity of large annotated datasets in Bengali further affects generalization, as deep learning models typically require substantial amounts of labeled data for robust performance. While our proposed Fusion Transformer-XL model mitigates some of these challenges by leveraging BiGRU and BiLSTM layers, which enhance context awareness and long-range dependency handling, class imbalances and subtle semantic variations between cyberbullying categories still pose limitations. Addressing these issues requires a more tailored approach, including advanced preprocessing techniques, adaptive sampling strategies, and better representation learning methods. In real-world applications, such as social media monitoring, content moderation, and automated cyberbullying detection systems, the ability to handle diverse linguistic structures is crucial for achieving accurate and fair results. The findings suggest that hybrid deep learning architectures can provide improved accuracy in low-resource languages, but further research is needed to enhance preprocessing methods, dataset expansion, and cross-lingual adaptability. Future work should explore domain adaptation techniques, semi-supervised learning, and improved tokenization for Bengali text processing to further refine cyberbullying detection in multilingual contexts.

## LIMITATION AND FUTURE WORK

It is crucial to acknowledge the limitations of the proposed model to enhance its effectiveness in future applications. A key limitation of this study is its reliance on specific datasets, as we utilize both English and Bengali datasets for this research. While this bilingual approach improves linguistic diversity, dataset dependency may still introduce biases and limit the model's ability to generalize across different linguistic styles and sociocultural contexts. This dependency could result in performance disparities when applied to other languages or dialects, particularly those with different syntactic structures,

informal expressions, or code-mixing patterns. Furthermore, potential biases in data collection, such as the underrepresentation of specific user demographics or cyberbullying patterns, may affect the model's fairness and robustness. Expanding training data to include more diverse sources and dialectical variations would improve generalizability and mitigate bias.

Another limitation is the availability and quality of labeled data, which directly impacts the performance of the Fusion Transformer-XL model. Low-resource languages, such as Bengali, often lack extensive annotated datasets, making it challenging to develop robust deep-learning models. Addressing this limitation requires leveraging semi-supervised learning techniques, data augmentation, or transfer learning from high-resource languages to improve performance in multilingual settings. Additionally, our study primarily focuses on linguistic features for cyberbullying detection, neglecting other contextual information such as user metadata and multimodal content (*e.g.*, images, videos, emojis). Future research should explore multimodal methodologies that integrate textual, visual, and behavioral indicators to enhance cyberbullying detection across various online platforms.

To further refine the applicability of our approach, future work should also investigate model interpretability and explainability, ensuring transparency in decision-making processes. Developing explainable AI techniques would enable real-world deployment in social media moderation systems, fostering ethical and fair applications of cyberbullying detection models.

## CONCLUSION

This work introduces a thorough method for identifying instances of cyberbullying in Bengali writing by using a Fusion Transformer-XL model. The proposed model utilizes sophisticated approaches such as relative positional encoding and incorporates both bidirectional GRU and LSTM layers to capture contextual intricacies and long-term interdependencies in textual input accurately. The model exhibits strong and consistent performance across several measures by undergoing intensive training and optimization, including techniques such as early stopping and k-fold cross-validation. A comprehensive evaluation of the effectiveness of the model is achieved by including metrics such as accuracy, recall, F1-score, confusion matrices, and ROC curves. The findings demonstrate that the hybrid Transformer-XL model not only enhances the precision of cyberbullying detection but also provides a flexible solution that can be customized for different languages and situations. Despite these limitations, this study proposes a novel method of detecting cyberbullying in Bangla and establishes a foundation for future research to expand upon these discoveries.

### Funding

This research is funded by the Research Chair of Online Dialogue and Cultural Communication at King Saud University, Riyadh, Saudi Arabia. The funders had no role

in study design, data collection and analysis, decision to publish, or preparation of the manuscript.

## Grant Disclosures
The following grant information was disclosed by the authors:
Research Chair of Online Dialogue and Cultural Communication at King Saud University, Riyadh, Saudi Arabia.

## Competing Interests
The authors declare that they have no competing interests.

## Author Contributions
- Md. Mithun Hossain conceived and designed the experiments, performed the computation work, prepared figures and/or tables, and approved the final draft.
- Md. Shakil Hossain conceived and designed the experiments, performed the computation work, prepared figures and/or tables, and approved the final draft.
- Md. Shakhawat Hossain conceived and designed the experiments, analyzed the data, performed the computation work, prepared figures and/or tables, and approved the final draft.
- M. Firoz Mridha conceived and designed the experiments, authored or reviewed drafts of the article, and approved the final draft.
- Mejdl Safran performed the experiments, analyzed the data, authored or reviewed drafts of the article, and approved the final draft.
- Sultan Alfarhood performed the experiments, analyzed the data, performed the computation work, authored or reviewed drafts of the article, and approved the final draft.
- Dunren Che performed the experiments, authored or reviewed drafts of the article, and approved the final draft.

## Data Availability
The code is available at GitHub and Zenodo:

- https://github.com/MIthun667/Fusing-Transformer-XL-for-Cyberbullying-Detection/tree/main.

- Hossain, M. M. (2025). Cyberbullying Detection. In PeerJ Computer Science. Zenodo. https://doi.org/10.5281/zenodo.15004841.

The Cyberbullying Detection Dataset in English is available at Kaggle and Zenodo:

- https://www.kaggle.com/datasets/momo12341234/cyberbully-detection-dataset.

- Hossain, M. M. (2025). Cyberbullying Detection [Data set]. Zenodo. https://doi.org/10.5281/zenodo.15004873.

The Cyberbullying Detection Dataset in Bengali is available at Kaggle: https://www.kaggle.com/datasets/cypher1337/dataset-for-cyberbully-detection-bengali-comments.

## Supplemental Information

Supplemental information for this article can be found online at http://dx.doi.org/10.7717/peerj-cs.2940#supplemental-information.

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
