# Peer review of "Fusing Transformer-XL with bi-directional recurrent networks for cyberbullying detection"

_PeerJ Computer Science, doi:10.7717/peerj-cs.2940_

## Round 0.1 · original submission · Minor Revisions

Please revise the paper following reviewer suggestions.

·

Basic reporting

(1) In the abstract, the proposal, method, and main results are clearly outlined.

(2) In general terms, the paper is generally well-written, but there are a few areas where the grammar can be improved for better clarity and readability. ​

(*) In proposed Methodology Section: ​
Line 169: Figure 1 shows proposed Fusion Transformer-XL model architecture uses a combination of Transformer-XL, Bidirectional Gated Recurrent Units (BiGRU), and Bidirectional Long Short-Term Memory (BiLSTM) networks to address the problem of cyberbullying detection. ​
Suggestion: Figure 1 shows the proposed Fusion Transformer-XL model architecture, which uses a combination of Transformer-XL, Bidirectional Gated Recurrent Units (BiGRU), and Bidirectional Long Short-Term Memory (BiLSTM) networks to address the problem of cyberbullying detection. ​

(*) In discussion Section:
Line 574: In this part, we explore the consequences of the results and the wider framework of the research. ​
Suggestion: In this section, we explore the implications of the results and the broader context of the research. ​

(3) In the RESULT ANALYSIS section there may be redundant information in tables and graphs, I would opt for tables.

(4) The similar content of the paper is around 4%, especially similarities of words and phrases, so it is possible to indicate that it is unique content.

Experimental design

The experimental design is concise and clear to demonstrate the working hypothesis.

Validity of the findings

In the literature review and table 9 (comparison) of the discussion section consider adding more recent studies using transformer-based approach as below (and others):

@article{SIHABUSSAKIB2024100104,
title = {Cyberbullying detection of resource constrained language from social media using transformer-based approach},
journal = {Natural Language Processing Journal},
volume = {9},
pages = {100104},
year = {2024},
issn={2949-7191},
doi = {https://doi.org/10.1016/j.nlp.2024.100104},
url = {https://www.sciencedirect.com/science/article/pii/S2949719124000529},
author = {Syed Sihab-Us-Sakib and Md. Rashadur Rahman and Md. Shafiul Alam Forhad and Md. Atiq Aziz},
}

@INPROCEEDINGS{10630719,
author={Tapaopong, Wachiraporn and Charoenphon, Atiphan and Raksasri, Jakkapong and Samanchuen, Taweesak},
booktitle={2024 5th Technology Innovation Management and Engineering Science International Conference (TIMES-iCON)},
title={Enhancing Cyberbullying Detection on Social Media Using Transformer Models},
year={2024},
volume={},
number={},
pages={1-5},
keywords={Training;Analytical models;Accuracy;Transfer learning;Text categorization;Cyberbullying;Predictive models;cyberbullying detection;text classification;transformer models;transfer learning},
doi={10.1109/TIMES-iCON61890.2024.10630719}}

Additional comments

I congratulate the researchers for dedicating their time to improving coexistence and reducing practices that are harmful to society.

Reviewer 2 ·

Basic reporting

The paper is well-organised and written in clear English. It effectively introduces the issue of detecting cyberbullying in the Bengali language, highlighting a significant gap in existing research. The methods and referenced works are detailed adequately, ensuring the paper meets academic standards. However, the discussion could be expanded to better explain the impact of language-specific challenges on the model’s performance.

Experimental design

The experimental approach is thoroughly outlined, providing enough detail about the methods and data to allow others to replicate the study. The innovative use of combining Transformer-XL with bi-directional recurrent networks (BiGRU-BiLSTM) for cyberbullying detection is a strong aspect of the design. The paper clearly explains the data preparation steps such as cleaning, upsampling, and augmentation. However, it could benefit from a deeper explanation of why certain model parameters were chosen and how these choices affect the results.

Validity of the findings

The results are convincing, with high accuracy and F1 scores indicating that the model effectively detects cyberbullying. Cross-dataset evaluation further validates the findings, showing that the model is effective across various data sets. Nonetheless, the paper would be improved by a discussion on possible data biases and the limitations of applying the model to other languages or data sets. Such a discussion would help clarify the generalisability and reliability of the model in different settings.

---

## Round 0.2 · accepted · Accept

Thank you for carefully addressing reviewer concerns in this revision. Based on the reassessment of both reviewers, I can now recommend acceptance of this submission.

While one of the reviewers points out that the Kaggle link provided is broken, I have checked the two Kaggle links provided in the paper and both worked for me. I would still request authors to please double check that all the provided links work.

·

Basic reporting

Reviewing the corrections, the authors effectively address the questions and recommendations made.

I also reviewed the source code from Zenodo repository verifying its availability. I must point out that the link to the Kagle platform is broken, I did not have access to the dataset.

After solving that, in my opinion, the article could be accepted.

Regards

Experimental design

No-comments

Validity of the findings

No-comments

Additional comments

After solving a problem in URL to access the open dataset, the article could be accepted.

Reviewer 2 ·

Basic reporting

The authors have addressed all comments thoroughly and effectively.

Experimental design

The authors have addressed all comments thoroughly and effectively.

Validity of the findings

The authors have addressed all comments thoroughly and effectively.